# Do NSAIDs Really Interfere with Healing after Surgery?

**DOI:** 10.3390/jcm10112359

**Published:** 2021-05-27

**Authors:** Stephan A. Schug

**Affiliations:** Anaesthesiology and Pain Medicine, Medical School, University of Western Australia, 6000 Perth, Australia; stephan.schug@uwa.edu.au

**Keywords:** NSAID, postoperative analgesia, bone healing, soft tissue healing, wound healing, cartilage repair, anastomotic leakage

## Abstract

Perioperative analgesia should be multimodal to improve pain relief, reduce opioid use and thereby adverse effects impairing recovery. Non-steroidal anti-inflammatory drugs (NSAIDs) are an important non-opioid component of this approach. However, besides potential other adverse effects, there has been a longstanding discussion on the potentially harmful effects of NSAIDs on healing after surgery and trauma. This review describes current knowledge of the effects of NSAIDs on healing of bones, cartilage, soft tissue, wounds, flaps and enteral anastomoses. Overall, animal data suggest some potentially harmful effects, but are contradictory in most areas studied. Human data are limited and of poor quality; in particular, there are only very few good randomized controlled trials (RCTs), but many cohort studies with potential for significant confounding factors influencing the results. The limited human data available are not precluding the use of NSAIDs postoperatively, in particular, short-term for less than 2 weeks. However, well-designed and large RCTs are required to permit definitive answers.

## 1. Introduction

Current concepts of perioperative pain control emphasise the value of multimodal analgesia, defined as the combined use of multiple analgesic medications or techniques with different mechanisms or sites of action [1]. Multimodal analgesia improves postoperative pain control and has an opioid-sparing effect. This leads to reduced rates of opioid-related adverse effects such as nausea, vomiting and constipation and thereby has the potential to speed up and improve postoperative recovery. Reducing opioid use after surgery and trauma is also a desirable outcome in view of the current issues with over prescription of opioids in general and the increased risk of long-term opioid use by initiation of acute pain management with opioids [2]. The concept is supported by good evidence and strongly recommended by international guidelines [3,4].

However, the best combinations of medications and techniques used are still under investigation and may well need to be procedure-specific [5]. A universal component of most multimodal analgesic approaches are non-steroidal anti-inflammatory drugs (NSAIDs). Ideally combined with paracetamol, they provide well-proven beneficial effects with regard to improved analgesia and reduced opioid requirement, as demonstrated in a network meta-analysis [6]. While paracetamol can be used nearly universally due to negligible adverse effects, NSAIDs carry a much higher risk of adverse effects, limiting their use [7]. The introduction of cyclooxygenase-2 (COX-2) selective NSAIDs (coxibs) has reduced these risks with regard to gastrointestinal ulcers, bleeding complication and bronchospasm [4]. Coxibs may even carry a lower risk than some nonselective NSAIDs (nsNSAIDs) with regard to kidney compromise and a similar risk of cardiovascular events [8]; in a meta-analysis, the injectable coxib parecoxib used perioperatively had no statistically different risk than for all adverse effects [9].

One issue which continues to be discussed in this context is the effect of NSAIDs (nsNSAIDs and coxibs) on postoperative and posttraumatic healing of soft tissues, bones and anastomoses, here with the potential risk of anastomotic leakage. This paper will review the current evidence on this contentious question.

## 2. Results

### 2.1. Bone Healing

Bone tissue undergoes a continuous remodelling process involving ongoing resorption, formation and mineralisation. Prostaglandins, in particular prostaglandin E2 (PGE2), are involved in maintaining this process and essential to the balance of osteoblasts and osteoclasts [10]. Bone healing after a fracture is different from healing of other injuries, in so far as it is a process of regeneration of normal bone tissue and not formation of a scar. It is therefore the result of the normal remodelling process and thereby dependent on prostaglandins [11]. This has resulted in theoretical concerns about the effects of NSAIDs on bone healing, as their main effect is inhibition of prostaglandin synthesis. These concerns may in particular relate to coxibs, as COX-2 is the dominant cyclo-oxygenase in osteoblasts.

In line with these physiologically founded concerns, an initial study of indomethacin use after closed femoral fracture in a rat model showed delays in fracture healing and formation of inferior callus predisposing to non-union [12]. This triggered a discussion on the potentially harmful effects of NSAIDs on bone healing, which is still ongoing. A number of subsequent animal studies, mainly in rodents, confirmed the detrimental effect of NSAIDs, in particular coxibs, on early fracture repair and the subsequent bone density and strength [13]. However, these effects are not consistently reported and translation to demonstrable effects on fracture healing in clinical settings in human patients is even less convincing.

A most recent meta-analysis of effects of NSAIDs in humans shows an association between NSAID exposure and delayed union or non-union (OR 2.07; 95% CI 1.19 to 3.61) [14]. However, the authors note the lack of randomised controlled trials (RCTs) and an increased odds ratio in studies of low quality. Confounding factors in the observational cohort studies such as patients using more NSAIDs for longer due to increased pain for a poorly healing fracture could have contributed to these results. Furthermore, there was no association shown with low dose or short-term use (<2 weeks) (OR 1.68; 95% CI 0.63 to 4.46) or in paediatric patients (OR 0.58; 95% CI 0.27 to 1.21). A preceding qualitative systematic review could not pool data for a meta-analysis, criticised the low study quality and concluded that there is no strong evidence that NSAIDs used for pain relief lead to increased rates of non-union [15]. However, the authors state that due to poor and conflicting data, no clinical recommendations can be made. These conclusions are in line with those of another older evidence-based review [16].

The concerns about low study quality showing an increased odds ratio for non-union are a repeating topic in the literature. A preceding meta-analysis showed no increased risk of non-union when only based on the highest quality studies [17]. A deliberate attempt to correlate research quality and results in this area in a systematic review has been undertaken [18]. Assessing the quality of the clinical studies using a modified Coleman Methodology Score, it was found that this score was significantly lower (*p* = 0.032) for those studies showing harmful effects on bone healing (40.0 +/− 14.3 points vs. 58.8 +/− 10.3 points). Observational studies are difficult to interpret and there may be unidentified confounding factors. This is illustrated by a study based on prescription database analysis, in which non-union after long bone fracture was increased by coxibs (OR 1.84; 95% CI 1.38 to 2.46) and surprisingly to a similar extent by opioids (OR 1.69; 95% CI 1.53 to 1.86), but not by non-selective NSAIDs (OR 1.07; 95% CI 0.93 to 1.23) [19].

Studies published subsequent to the latest meta-analysis are suggesting little effects of NSAIDs on bone healing. In a recent small RCT (*n* = 128) of open reduction and internal fixation of ankle fractures, ketorolac vs. no ketorolac had no effect on clinical or radiographic healing assessed by blinded observers at 12 weeks postoperatively [20]. A randomized controlled trial (RCT) in patients with Colles’ fracture (*n* = 95) showed no difference in outcome between the groups receiving ibuprofen and placebo [21]. In another more recent multicenter observational study, NSAID use versus opioid use after intermedullary nailing of diaphyseal tibia fractures (*n* = 372) had no effect on healing time [22]. The lack of an effect in pediatric patients has also been confirmed in an RCT of NSAID (ibuprofen) versus no NSAID (*n* = 102) after long bone fractures; there were no differences in time to clinical or radiographic healing [23].

Even studies undertaken in patients undergoing operations where bone healing or fusion is a desired outcome of surgery do not show a detrimental effect of NSAIDs on this surgical outcome. After osteotomy of a long bone, there was no difference in the time to union between patients treated with an NSAID pain protocol or an NSAID-free pain protocol [24]. After spinal fusion, a systematic review could not identify any RCT and concluded from the studies of lower quality, that postoperative NSAID short-term use (for less than 2 weeks) does not increase the risk of non-union [25].

For bone healing overall, there is some evidence from animal studies that, in particular, COX-2 inhibition may suppress fracture healing, although even this is not consistent. In human patients, there is no robust evidence to support this concern. On the basis of the available human data and in view of the limited quality of these, there is currently no reason to deny patients after a fracture pain relief by short-term use of nsNSAIDs and most likely coxibs.

### 2.2. Cartilage Repair

Similar to the concerns about bone healing, there has been an ongoing discussion on the effects of NSAIDs on chondrocytes. In vitro and animal studies report early negative effects on chondrocyte proliferation, but not at later time points, and on chondrocyte differentiation, but again these data are not consistent [26]. This systematic review claims to have identified only one human study; however, this is a study of anterior cruciate ligament repair and will be discussed below [27].

Overall, the conclusions are that it is currently unclear, due to a lack of any studies in humans, what effects NSAIDs have on the outcome of cartilage repair procedure.

### 2.3. Musculoskeletal Soft Tissue Healing (Tendons and Ligaments)

A systematic review of animal studies on tendon-to-bone healing was compromised by heterogeneous outcome reporting [28]; a subgroup analysis of homogenous animal studies showed no effect by NSAIDs.

However, in a rabbit model of rotator cuff repair, nsNSAIDs had no (ibuprofen) or only minimal early (flurbiprofen) effects on healing, while coxibs (celecoxib) resulted in lower failure loads at any time point between 3 and 12 weeks [29]. Similar results were found in a human RCT (*n* = 180) with higher re-tear rates after use of celecoxib versus ibuprofen and tramadol [30].

The above study was also the only RCT included in a meta-analysis of studies which looked at effects of NSAIDs on soft tissue healing in general [31]. It analysed another 3 observational studies, of which the largest contributed 93% of cases (anterior cruciate ligament repair); it found overall no difference in surgical failure rate and other outcomes between NSAID and no NSAID use (3.6 vs. 3.7%).

There are insufficient human data on the effect of musculoskeletal soft tissue healing, but their use may not have a detrimental effect, possibly with the exception of celecoxib use after rotator cuff repair.

### 2.4. Free Flaps

In an animal study in rats, perioperative celecoxib use had no harmful effects on flap survival and healing compared to non-use [32].

In a retrospective cohort study (*n* = 138), ketorolac use after head and neck free flaps did not increase bleeding complications or risk of free flap failure [33].

With regard to coxibs, the results are contradictory. An animal study in rats showed a detrimental effect on flap survival with use of parecoxib [34], but not with celecoxib [32]. The only human study here is a cohort study with historical controls with the associated risk of potential confounding factors [35]. The introduction of parecoxib and valdecoxib for two to three weeks postoperatively resulted in an increase of free flap failure from a base line rate of 7% to 29%, which fell to 4% when coxibs were no longer used.

### 2.5. Anastomotic Leakage

Use of NSAIDs after gastrointestinal surgery is of particular interest, as NSAIDs improve recovery of gastrointestinal function (faster return of flatus, stool passage and oral feeding tolerance) and would therefore enhance postoperative rehabilitation [36].

However, there is an ongoing discussion on the potential of NSAIDs to increase anastomotic leakage. This concern has been partially driven by rodent data showing reduced collagen formation in subcutaneous granulation tissue due to exposure to diclofenac, although anastomotic and skin wound strength were not affected in this study [37]. Two most recent systematic reviews of primarily cohort studies with an overlap of four studies showed an increase in anastomotic leakage rate with all nsNSAIDs; a later smaller one including 8 studies found an OR of 1.79 (95% CI 1.47 to 2.18) [38] and the bigger one including 17 studies an OR of 2.02 (95% CI 1.62 to 2.50) [39]. In the latter meta-analysis, there was no difference if only RCTs were analysed, suggesting a potential risk of bias by basing these systematic reviews to a large extent on cohort studies [39]. It is in particular of note, that in both these systematic reviews, subgroup analysis found no increased risk with selective COX-2 inhibitors.

Overall, healing of gastrointestinal anastomoses is unlikely to be impaired by perioperative use of NSAIDs. This is better documented for coxibs than for nsNSAIDs.

### 2.6. Wound Healing

Animal studies on wound healing report contradictory results. For example, specifically for coxib use, there are studies which show impaired healing [40], no effect on healing [41] or improved healing [42]. There are no published human data on this issue.

## 3. Discussion

The question as to whether NSAIDs really interfere with healing after surgery or trauma cannot be answered definitively on the basis of the currently published literature. This is primarily due to contradictory results of animal experiments and the lack of properly designed RCTs in this setting. The issues with the published cohort studies, often with historical controls, are the significant risk of confounding factors influencing the results. This has been shown in particular for bone healing, where harmful effects of NSAIDs are, in particular, found in studies of poor quality. Therefore, it is of utmost importance that properly designed RCTs are performed to address the questions related to this important issue.

In the interim, the data currently available should not preclude the use of NSAIDs postoperatively, in particular, when used in the short term for less than 2 weeks. Overall, data suggest that coxibs are safer in the setting of GI anastomoses, but may have more negative effects on bone and soft tissue healing, but again this statement is based on very poor data and may be speculative.

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
