# Peer review of "Do NSAIDs Really Interfere with Healing after Surgery?"

_jcm, 2021, doi:10.3390/jcm10112359_

Round 1

Reviewer 1 Report

The author has reviewed the effects of NSAID's on healing after surgery. The review focuses on bone healing after surgery or trauma, only short paragraphs describe the effect on cartilage repair, musculoskeletal healing, free flaps, anastomotic healing and wound healing. 

Major concerns:

  1. Unfortunately, hardly any attention is paid to the pathophysiological background.
  2. Reads unpleasant as it is mainly a summary of various studies
  3. The review is hardly innovative. the review is hardly innovative.de recensie is nauwelijks innovatief.

Author Response

Re lack of pathophysiological background: each paragraph mentions briefly the potential pathophysiology and quotes relevant reviews on the pathophysiology or relevant basic science studies.

Re style and lack of innovative ideas – the topic of the review was based on an invitation by the editors of the special issue – my review quotes the latest suitable publications on this and I have aimed to follow the instructions for authors of JCM, which suggests for reviews to ‘provide concise and precise updates on the latest progress made in a given area of research.’

Reviewer 2 Report

This narrative review by Professor Schug presents data on NSAID use and the potential interference with soft-tissue, bone, wound and anastomosis healing. The review is relevant to most clinicians and stimulates the reader to further examine and evaluate the potential of NSAIDs in the perioperative pain management. The paper is highly relevant, as pain management is an important subject for all physicians, especially with the alarming opioid overprescription and overuse around the world. Prof. Schug points the relevance of the paper very well in the beginning of the manuscript, leaving no doubt about the subject's importance to the reader. The introduction is spot-on. 

The manuscript is divided in different subsections in a way that makes good sense. There is a good flow in the manuscript, starting with bone healing in which there is most data on NSAID use and ends with wound healing and NSAIDs of which there is little to no strong evidence to discuss. The discussion section is short but includes and emphasizes the most important aspects regarding NSAID use and healing. Furthermore, the author highlights the urgent need for specifically randomized controlled trials on NSAID use and healing, as the dilemma with cohort studies/observational studies are finely and concisely pointed. 

In general, the paper is well written, concise and to-the-point. The English language, style and grammar are spotless.

This reviewer was engulfed into the manuscript and is just a little bit disappointed about the length of the manuscript. 

Author Response

Thanks for this positive review - re the comment about length I attempted to follow the instructions for authors which suggest that reviews are 'concise and precise updates'! 

Reviewer 3 Report

This is a comprehensive review on the effects of NSAID´s on soft tissue healing. The data are well presented. Provides a good overview about the animal and human data on this topic. If possible, elaborate more on the topic of NSAIDS and gastrointestinal anastomosis leakage. 

Author Response

Thanks for this positive review.

As suggested I have extended the paragraph on anastomosesotic leakage in the revised manuscript.